# Impact of the COVID-19 Pandemic on Epidemiological Trends in Pediatric Cervical Abscess-Forming Infections

**DOI:** 10.3390/microorganisms13010190

**Published:** 2025-01-17

**Authors:** Shuhei Takahashi, Ai Kishino, Kentaro Miyai, Shigeru Takishima, Tae Omori, Hidehiro Furuno, Ryosei Iemura, Makoto Ono, Keisuke Ogasawara, Akito Sutani, Masayuki Nagasawa

**Affiliations:** 1Department of Pediatrics, Musashino Red Cross Hospital, 1-26-1, Kyonan-cho, Musashino-City, Tokyo 180-8610, Japan; shu.takahashi.1204@gmail.com; 2Department of Pediatrics and Developmental Biology, Institute of Science Tokyo, 1-5-45 Bunkyo-ku, Tokyo 113-8519, Japan; 3Department of Pediatrics, Tokyo Bay Urayasu-Ichikawa Medical Center, 3-4-32 Todaijima, Urayasu-City, Chiba 279-0001, Japan; aiki@jadecom.jp; 4Department of Pediatrics, Tokyo Kita Medical Center, 4-17-56 Akabanedai, Kita-ku, Tokyo 115-0053, Japan; kmiyai.ped@gmail.com; 5Department of Pediatrics, Soka Municipal Hospital, 2-21-1 Soka, Soka-City, Saitama 340-8560, Japan; shigeru.takishima@gmail.com; 6Department of Pediatrics, Bokutoh Hospital, 4-23-15 Kotobashi, Sumida-ku, Tokyo 130-8575, Japan; qwt04303@nifty.ne.jp; 7Department of Pediatrics, Chiba Kaihin Municipal Hospital, 3-31-1 Isobe, Mihama-ku, Chiba-City, Chiba 261-0012, Japan; hidehiro.furuno@gmail.com (H.F.); a.iemura.814246@gmail.com (R.I.); mono-endo@umin.ac.jp (M.O.); 8Department of Pediatrics, Tuchiura Kyodo General Hospital, 4-1-1 Otsuno, Tsuchiura-City, Ibaraki 300-0028, Japan; guts2keiogasawara@gmail.com; 9Department of Pediatrics, Kawaguchi Municipal Medical Center, 180 Nishiaraijuku, Kawaguchi-City, Saitama 333-0833, Japan; stnakt@gmail.com

**Keywords:** COVID-19 pandemic, deep neck abscess, retropharyngeal abscess, peritonsillar abscess, children, cervical abscess

## Abstract

Abscess-forming cervical bacterial infections are rare and serious infections. **Methods:** We retrospectively examined the trends in abscess-forming cervical bacterial infections in children who required inpatient treatment in three periods before (January 2016 to June 2020), during (July 2020 to December 2022) and after the COVID-19 pandemic (January 2023 to June 2024). **Results:** The study included 96 patients with superficial cervical abscesses and 111 patients with deep cervical abscesses (34 with retropharyngeal abscesses, 51 with peritonsillar abscesses, and 26 with deep neck abscesses). Both decreased during the COVID-19 pandemic and increased significantly after the COVID-19 pandemic compared to before the COVID-19 pandemic (0.94 ± 0.92 vs. 0.50 ± 0.72 vs. 1.67 ± 1.11/month, 0.93 ± 0.96 vs. 0.60 ± 0.84 vs. 2.39 ± 1.70/month), which was related with the trends of respiratory viral infections. Bacteria were identified in 79 of the 97 cases in which punctures were performed; however, there were no significant differences between the three periods. No significant changes were found in the pharyngeal streptococcal antigen positivity rate, rate of oral antibiotic use before hospitalization, length of hospital stay, or duration of antibiotic administration before and after the COVID-19 pandemic. **Conclusions:** The COVID-19 pandemic has affected the epidemiology of cervical abscess-forming bacterial infections in children. Although the reemergence of respiratory viral infections after the COVID-19 pandemic may be a factor, the cause of the doubling in the number of neck abscesses after the COVID-19 pandemic remains unclear and requires further investigation.

## 1. Introduction

Deep neck bacterial infection is a general term for bacterial infections that occur within the spaces formed by loose connective tissue in the head and neck region, including lymphadenitis, cellulitis, and abscesses. It is speculated that this depends on the anatomical characteristics of the infected area, the nature of the causative bacteria, and the strength of the host defense mechanism, and whether a neck infection develops into an abscess or cellulitis [1]. Once an abscess forms, it is an emergency disease that can easily spread to the skull base or mediastinum and must be treated as a severe bacterial infection upon diagnosis [1].

Surgical treatment is the mainstay of treatment [2]. Recently, the number of successful cases of conservative treatment using antibiotics alone has been increasing, although puncture and drainage are still important treatment methods, not only for curative treatment, but also for identifying the causative bacteria to select appropriate antibiotics [3,4,5]. Furthermore, it is an essential emergency treatment for airway obstruction caused by swollen abscesses. It is extremely difficult to clinically judge the appropriateness and timing of these procedures in children because of their anatomical peculiarities and technical difficulties [1,6].

As has been widely reported, the COVID-19 pandemic not only has a socioeconomic impact, but also has a major impact on endemic infectious diseases [7,8,9]. Due to the spread of non-pharmaceutical interventions (NPIs), such as handwashing and mask-wearing, many epidemic infectious diseases have significantly decreased and have gradually recovered [10,11,12].

Against this background, we retrospectively investigated the trends in abscess-forming neck infections in children, which are relatively rare and serious infections, as a multicenter collaborative study.

## 2. Materials and Methods

The data of patients aged < 15 years were extracted from the medical chart records of patients admitted between January 2016 and June 2024 using the following keywords: lymphadenitis, abscess, peritonsillar abscess, retropharyngeal abscess, and deep neck abscess. In accordance with the general practice guidelines [13], each facility actively performed contrast-enhanced computed tomography (CT) scans on cases with Red Flag signs such as neck pain and dysphagia to advance the diagnosis. Patients in whom ring enhancement and abscess formation were confirmed in the neck region using contrast-enhanced CT were registered. The final diagnosis was based on contrast-enhanced CT findings of the physician and facility radiologist. For patients transferred before discharge, we were unable to obtain detailed information on the duration of antibiotic treatment or the length of hospitalization in most cases, including whether surgical drainage was performed. These cases were excluded from the analysis owing to missing information.

Empiric antimicrobial therapy was continued if there was clear improvement in clinical symptoms such as fever resolution, improvement in sore throat, or improvement in neck pain within 48–72 h of initiation of intravenous antimicrobial therapy [3,14]. Switching from intravenous antimicrobial therapy to oral antimicrobial therapy was determined on a case-by-case basis based on apparent reduction in abscess size and normalization of inflammatory response by blood test. Steroids were used intravenously as needed at the discretion of the physician to prevent edema of the abscess [15].

Patients were recruited from eight core regional hospitals in and around the Tokyo metropolitan area: Musashino Red Cross Hospital, Tokyo Bay Urayasu-Ichikawa Medical Center, Tokyo Kita Medical Center, Soka Municipal Hospital, Bokutoh Hospital, Chiba Kaihin Municipal Hospital, Tuchiura Kyodo General Hospital, and Kawaguchi Municipal Medical Center. The locations of the eight participating facilities are shown in Figure 1. According to the administrative data in Japan (https://www.stat.go.jp/data/jinsui/, accessed on 26 December 2024), the total population of the region where the participating facilities are located increased by a factor of 1.014–1.049 (average 1.032) from 2016 to 2024, while the pediatric population declined by a factor of 0.760–1.075 (average 0.911) during the same period. The effect of fluctuations in the target pediatric population was considered negligible for the epidemiological interpretation.

The study was approved by the ethics committee of the Musashino Red Cross Hospital, the main research facility, and the ethics committee of each medical institution. Consent was obtained from all patients using the opt-out method.

### Statistical Analysis

The statistical testing utilized Chi-square analysis, correlation analysis, Student’s *t*-test, and Mann–Whitney U analysis were used for statistical analysis, and a *p* < 0.05 was determined as significant. Statistical analyses were performed using JMP 14 software (SAS Institute, Cary, NC, USA).

## 3. Results

A total of 96 patients with superficial abscess-forming infections and 111 patients with deep cervical abscess-forming infections were enrolled. Most of the former were cervical lymph node abscesses, with three abscesses in the median cervical cyst, five abscesses related to a piriform fistula, two parotid abscesses, and four submandibular abscesses. Of the latter, 34 were retropharyngeal abscesses, 51 were peritonsillar abscesses, and 26 were deep neck abscesses. None of the enrolled patients required emergency airway unblocking such as tracheotomy or endotracheal intubation.

Figure 2 shows the monthly frequency of these cases. During the COVID-19 pandemic, both superficial and deep cervical abscesses decreased by approximately 50%. From 2023 onwards, they increased significantly by approximately twice as much as before the COVID-19 pandemic. Even when comparing the individual diseases of retropharyngeal abscess, peritonsillar abscess, and deep neck abscess, the number of cases increased significantly after the COVID-19 pandemic compared to before the COVID-19 pandemic.

Figure 3 shows the age distribution, and there was a tendency for superficial abscess-forming infections to occur at a younger age than deep cervical abscesses.

Next, we compared the clinical information before and after the pandemic. Among the registered patients, 22 were transferred to a high-quality hospital for surgical procedures during admission. In most cases, the necessary clinical information for the duration of antibiotic use and hospital stay could not be obtained. They included 8 cases of superficial cervical abscess and 14 cases of deep cervical abscess, of which 10 occurred before the pandemic, 5 cases were during the COVID-19 pandemic, and 7 cases were after the COVID-19 pandemic. In total, 2 patients underwent puncture drainage, 3 were treated conservatively with antibiotics without puncture drainage, and the presence or absence of puncture drainage was unknown in the remaining 17 patients.

Table 1 shows the frequency of occurrence in each period, average age, antibiotic administration before hospitalization, frequency and positivity rate of the pharyngeal streptococcal antigen test, number of cases in which punctures were performed, frequency of detected pathogens, number of cases in which steroids were used, duration of antibiotic treatment, and hospitalization. Superficial and deep cervical abscess-forming infection decreased during the COVID-19 pandemic and increased significantly after the COVID-19 pandemic compared to before the COVID-19 pandemic (0.94 ± 0.92 vs. 0.50 ± 0.72 vs. 1.67 ± 1.11/month, 0.93 ± 0.96 vs. 0.60 ± 0.84 vs. 2.39 ± 1.70/month, *p*-value < 0.05, Student’s *t*-test). Abscess punctures tend to be performed more often for superficial abscesses than deep cervical abscesses (odds: 1.64, 95% CI: 1.01–2.70, *p*-value = 0.02), and superficial abscess punctures were performed more before than after the COVID-19 pandemic (odds: 2.78, 95% CI: 1.15–6.67, *p*-value = 0.02). There was no significant difference in the frequency of punctures for deep cervical abscesses before and after the COVID-19 pandemic (odds: 0.80, 95% CI: 0.35–1.80, *p*-value = 0.58). Bacteria were identified in 79 of the 97 cases in which punctures were performed (Table 2). Of the 91 cases in which puncture drainage was performed, 47 (51.6%) were performed on the first day of admission and 74 (81.3%) within 3 days. The latest puncture drainage was performed on the 12th day of admission for two cases. In the two cases in which methicillin-resistant *Staphylococcus aureus* (MRSA) was detected, ampicillin/sulbactam (ABPC/sbt) was changed to vancomycin (VCM), which was administered for one week, followed by sulfamethoxazole/trimethoprim (ST combination), or ST combination plus rifampicin was administered orally for another week. Although statistical comparisons were difficult because of the small number of comparable bacterial species detected, there was no significant tendency for specific bacterial species to increase after the pandemic. Blood cultures were performed in 94 of 96 patients with superficial neck abscesses and 82 of 111 patients with deep neck abscesses, none of which were positive. No significant changes in average age, antibiotic administration before hospitalization, frequency and positivity rate of the pharyngeal streptococcal antigen test, number of cases in which steroids were used, duration of antibiotic treatment, or hospitalization were observed before and after the pandemic.

Table 3 shows the differences in the clinical course depending on the presence or absence of a puncture. The length of hospital stay tended to be shorter in the puncture group for deep cervical abscesses, whereas it was significantly longer in the puncture group for superficial neck abscesses. This is thought to be due to the fact that puncture was selectively performed for severe cases of superficial neck abscesses. For superficial and deep cervical abscesses, there was no significant difference in the total duration of antibiotic treatment between patients with and without punctures.

## 4. Discussion

This study found that the incidence of abscess-forming cervical infections decreased during the COVID-19 pandemic as well as in many other epidemic infectious diseases. Pediatric deep cervical abscesses are known to occur as complications of upper respiratory infections, and a reduction in respiratory infections may have contributed to these results [16]. Additionally, the incidence of superficial and deep cervical abscesses has increased since the pandemic. This may be attributed to an outbreak of upper respiratory tract infections following the COVID-19 pandemic. Respiratory viral infections, such as respiratory syncytial virus (RSV), also showed a significant decline in 2020; however, outbreaks began to recover as early as 2021. In 2023, adenovirus infections will cause the largest outbreak in the past decade [17]. It has been reported that viral ‘colds’ predispose to bacterial rhinosinusitis in both adults and children [18,19] and are among the most common infections seen in primary care [20]. In one prospective longitudinal study in children, 8% of viral upper respiratory tract infection (RTI) were complicated by acute bacterial rhinosinusitis [19]. It has also been demonstrated that there is a positive association between viral RTI and bacterial superinfection in rhinitis [21,22], RSV-induced bronchiolitis [23], and acute expiratory wheezing [24].

Several mechanisms are known as to how respiratory viral infections cause secondary respiratory tract bacterial infections. It has been shown that viral infections promote bacterial adherence and airway colonization via activation of TGF-β which induces upregulation of expression of fibronectin and integrins to which bacteria bind [25]. RSV induces upregulated expression of the cell surface glycoprotein intercellular adhesion molecule 1 (ICAM-1) by primary respiratory tract epithelial cells and that ICAM1 serves as a cognate ligand for *Haemophilus influenza* [26]. Virus infection induced dysregulation of the proinflammatory cytokine response such as overproduction of immunosuppressive IL-10, is generally believed to play a major role in predisposing to secondary bacterial infection [27,28]. Consequently, it will induce reduced function of dendritic cells, macrophages, natural killer cells, CD4+ and CD8+ T-cells, all of which leads to impaired ability to effectively eradicate bacterial co-pathogens [28]. No confirmatory studies have been reported on whether a similar mechanism applies to the relationship between respiratory viral infections and cervical abscess-forming infections so far. Nasopharyngeal viral infections can lead to secondary abscess-forming cervical infections.

In contrast, pneumococcal infections in children and invasive pneumococcal infections in the elderly are triggered by respiratory viral infections, such as the influenza virus [29,30,31]. According to epidemiological data from the Tokyo Metropolitan Government, the number of invasive pneumococcal infections in both children and adults has decreased to one-third of the pre-COVID-19 pandemic level and has gradually recovered since then, but remains at about two-thirds of the pre-COVID-19 pandemic level in the first half of 2024 [17]. It is unclear whether the resurgence of respiratory viral infections alone can explain the more than two-fold increase in abscess-forming neck infections compared with the pre-COVID-19 pandemic period.

Host immune responses are thought to be involved in the occurrence of abscess-forming infections [32]. It would be interesting to explore how changes in social activities caused by the three-year pandemic affect children’s immune development. There are various reports on the impact of immunological debt caused by a drastic reduction in epidemic infectious diseases due to the penetration of NPIs in pediatric infectious diseases after the pandemic [10,12,33,34]; however, its pathogenesis remains unknown and requires further research.

Regarding the pathogenic factors, no notable changes were observed in the causative bacteria before and after the COVID-19 pandemic. No notable changes were observed in the detection rates of *Streptococcus pyogenes* from the patient’s pharynx in our cohort. Since the second half of 2023, the number of fulminant streptococcal infections caused by the highly virulent M1uk strain has rapidly increased in Japan, mainly among adults [35], similar to other countries [36,37,38]. According to data on the strains of pharyngeal *Streptococcus pyogenes* monitored by the Tokyo Institute of Health, the frequency of the T1 serum strain to which the M1uk strain belongs did not increase in children from 2023 to 2024 [17]. Further detailed epidemiological and bacteriological analysis is needed to determine the involvement of M1uk strains in the increase in cervical abscess-forming infections. Panton–Valentine leukocidin (PVL)-producing organisms with strong tissue destructive properties are increasing in community-acquired methicillin-resistant *Staphylococcus aureus*, causing abscess-forming infections [39,40]. However, as shown in Table 2, methicillin-resistant *Staphylococcus aureus* was detected in only two cases, suggesting that PVL-producing bacteria did not contribute to an increase in abscess-forming infections. According to JANIS (Japan Nosocomial Infections Surveillance) data (https://janis.mhlw.go.jp/report/kensa.html, accessed on 28 December 2024), from 2018 to 2023, the frequency of MRSA among all *Staphylococcus aureus* isolated from inpatients decreased from 47.8% to 44.0%, and the frequency of MRSA among all *Staphylococcus aureus* isolated from outpatients decreased from 30.1% to 29.3%. It is unlikely that the antimicrobial resistance in *Staphylococcus aureus* is increasing in Japan as a whole.

On the other hand, although rare, pediatric cases of invasive infections caused by PVL-producing methicillin-sensitive *Staphylocuccus aureus* (PVL-MSSA) have been reported in recent years [41]. Also, there has been a report of an outbreak of PVL-MSSA infection in a neonatal unit [42].

The emergence of PVL-producing community-acquired MRSA is diversified from country to country; the rate of which is >90% in the USA, it is moderate in India at >65%, Korea is 0.9%, Germany 30%, Singapore 11.6%, Turkey 4% and China 12.8% [43,44]. There are few reports on the frequency of PVL-producing bacteria among MSSA, but a report from Nepal presented that strains with the PVL gene accounted for 7% of MSSA detected in healthy individuals [45]. Our study could not examine the PVL gene, so future studies are needed.

In this study, most pathogenic bacteria identified were *Staphylococcus* and *Streptococcus*. These results are consistent with those of previous reports [6,46]. In our study, ampicillin/sulbactam, which is effective against methicillin-sensitive *Staphylococcus*, *Streptococcus*, and many anaerobes, was used in most cases as empirical therapy (Appendix A), and treatment was completed without changing the antibiotics in many cases. In our cohort, only two cases of methicillin-resistant *Staphylococcus aureus* (MRSA) (2.5%, 2/79) were detected as resistant bacteria. Although culture-guided antimicrobial therapy is regarded as the best treatment for deep neck abscesses, surgical intervention may sometimes be difficult depending on the location of the abscess and the age of the child [6,46].

Our study will be helpful in determining empirical antibiotics for neck abscess-forming infections in cases where pathogenic bacteria are not identified.

The appropriate duration of antimicrobial therapy for deep neck abscess-forming infections is not clearly defined. The duration of antimicrobial therapy is often determined on a case-by-case basis, as it is influenced by factors such as the size and location of the abscess, the pathogenic bacteria, the presence and timing of puncture drainage and the response to antimicrobial therapy [14,47]. It is considered that our report provides valuable information regarding the appropriate duration of antimicrobial therapy for uncomplicated cervical abscess-forming infections.

In our cohort, blood cultures were performed in 94 of 96 patients with superficial neck abscesses and 82 of 111 patients with deep neck abscesses, none of which were positive. There are few reports on the positivity rate of blood cultures in cervical abscess-forming infections and deep neck abscesses. A retrospective observational study of 132 patients with cervical lymphadenitis/neck abscess aged 2 to 15 years reported that blood cultures were positive in 5/132 (3.8%) patients [48]. One of the possible reasons for the absence of positive blood cultures in our cases may include a history of exposure to antimicrobials prior to collection of blood cultures (Table 1).

Our study had some limitations. First, the causative bacteria were investigated in fewer than half of the cases. In the patients examined, blood culture tests were all negative, and we could not identify pathogenic bacteria, particularly in patients with deep neck abscesses. Pathogenic bacteria were identified only in patients who underwent surgical intervention. Patients requiring surgical intervention were often relatively severe compared to those who did not undergo surgical intervention, which may introduce bias in the identification of pathogenic bacteria. Furthermore, detailed bacteriological analyses, including genetic studies, could not be performed on the bacteria identified as causative bacteria. This was a retrospective multicenter observational study, and the diagnostic and treatment protocols were not standardized. Decisions regarding the necessity of surgical intervention were made by physicians at each institute. Finally, 22 patients were transferred to a high-quality hospital during admission, making it impossible to obtain the necessary clinical information.

Although cervical abscess-forming infections are relatively rare, they are clinically critical. We believe that this study is significant because it examined a relatively large number of cases compared to previous reports and provides useful information for understanding the pathophysiology of this disease based on epidemiological changes before and after the COVID-19 pandemic.

## 5. Conclusions

Superficial and deep cervical abscess-forming infections decreased during the COVID-19 pandemic and then more than doubled dramatically compared with pre-pandemic levels. There were no significant changes in the duration of treatment, length of hospitalization, or attributable organisms. The re-emergence of respiratory viral infections after the pandemic may be a factor; however, further observational studies are needed to determine whether other factors are present.

## Figures and Tables

**Figure 1 microorganisms-13-00190-f001:**
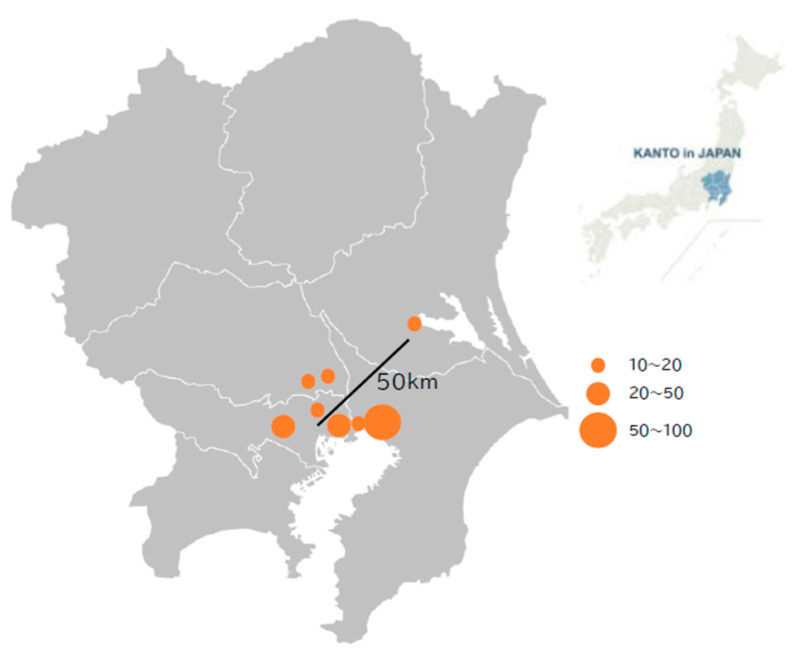
Location of research facilities. The Kanto region (**right**) in Japan and each facility location within the Kanto region (**left**) are presented. All but one facility are located within 25 km of central Tokyo. The circle size indicates the number of patients enrolled from each facility.

**Figure 2 microorganisms-13-00190-f002:**
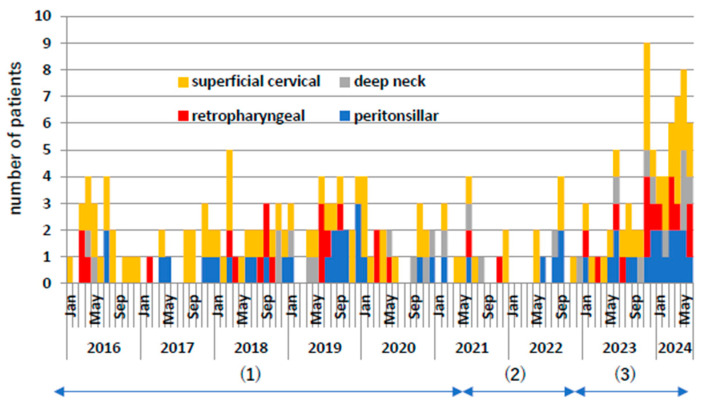
Number of patient occurrences per month. Yellow indicates superficial neck abscess, red indicates posterior pharyngeal abscess, gray indicates deep neck abscess, and blue indicates peritonsillar abscess. (1) Indicates pre-COVID-19 pandemic, (2) indicates during COVID-19 pandemic, and (3) indicates post-COVID-19 pandemic.

**Figure 3 microorganisms-13-00190-f003:**
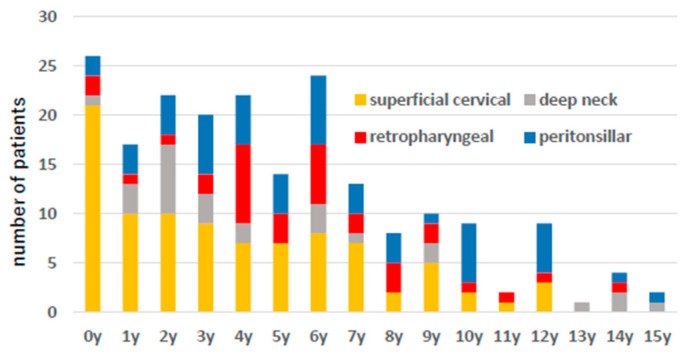
The number of patient cases by age is shown. Yellow indicates superficial neck abscess, red indicates posterior pharyngeal abscess, gray indicates deep neck abscess, and blue indicates peritonsillar abscess.

**Table 1 microorganisms-13-00190-t001:** Patients with superficial and deep cervical abscesses are summarized. Deep cervical abscesses are further divided into posterior pharyngeal abscesses, peritonsillar abscesses, and deep neck abscesses.

	n	n/Month	Age (y)	Antibiotics Use Prior to Admission	GAS Ag Positive (Test)	Puncture	Cultured Bacteria	Steroid Use	Antibiotics Therapy			Hospital Stay (Days)
									Iv (Days)	Po (Days)	iv + po (Days)	
**superficial cervical**	96	0.94 ± 0.92	3.71 ± 3.37	43	11(57)	**57**	48	1	9.1 ± 4.6	7.2 ± 5.1	15.8 ± 7.7	9.9 ± 4.3
① 2016.1–2020.6	51	0.96 ± 0.79	3.35 ± 3.20	24	2(28)	38	31	0	9.0 ± 3.6	7.6 ± 6.4	15.9 ± 8.9	9.7 ± 3.8
② 2020.7–2022.12	15	0.50 ± 0.72	2.93 ± 3.32	8	0(7)	11	11	0	9.9 ± 5.2	6.7 ± 2.9	16.6 ± 6.5	12.1 ± 5.9
③ 2023.1–2024.6	30	1.67 ± 1.11	4.72 ± 3.47	11	9(22)	8	6	1	9.2 ± 5.6	6.5 ± 2.5	15.0 ± 4.9	9.4 ± 3.2
**deep cervical**	111	1.09 ± 1.26	5.88 ± 3.77	35	30(58)	40	31	17	10.2 ± 5.3	7.8 ± 6.8	17.2 ± 9.2	10.7 ± 5.6
① 2016.1–2020.6	50	0.93 ± 0.96	5.12 ± 3.61	18	19(33)	19	16	9	11.0 ± 6.2	7.4 ± 5.5	17.3 ± 8.5	11.4 ± 6.1
② 2020.7–2022.12	18	0.60 ± 0.84	5.90 ± 4.83	4	1(7)	8	7	3	10.5 ± 5.5	8.9 ± 9.0	18.8 ± 11.0	11.0 ± 6.7
③ 2023.1–2024.6	43	2.39 ± 1.70	6.56 ± 3.24	13	10(18)	13	8	5	8.9 ± 3.8	7.8 ± 7.4	16.1 ± 9.4	9.7 ± 4.4
**retropharyngeal**	34	0.33 ± 0.66	5.74 ± 3.15	13	9(15)	13	8	8	11.8 ± 5.4	7.1 ± 4.1	18.4 ± 6.1	11.9 ± 5.6
① 2016.1–2020.6	18	0.33 ± 0.67	5.11 ± 3.03	7	7(11)	8	6	4	13.4 ± 6.2	6.4 ± 4.4	19.8 ± 6.4	13.2 ± 6.6
② 2020.7–2022.12	2			1	0(1)	0	0	0				
③ 2023.1–2024.6	14	0.78 ± 0.85	6.79 ± 2.81	5	2(3)	5	2	4	9.3 ± 3.3	6.2 ± 4.0	15.5 ± 4.3	9.8 ± 3.4
**peritonsillar**	51	0.50 ± 0.72	6.16 ± 3.79	15	14(30)	20	17	6	7.8 ± 3.8	7.8 ± 7.4	14.6 ± 9.9	8.4 ± 4.4
① 2016.1–2020.6	24	0.44 ± 0.71	5.38 ± 3.57	8	8(18)	10	9	4	8.2 ± 3.8	7.1 ± 5.4	14.1 ± 7.3	8.9 ± 4.0
② 2020.7–2022.12	8	0.27 ± 0.51	6.86 ± 5.14	1	1(3)	2	2	1	6.3 ± 3.5	7.0 ± 5.8	13.3 ± 9.2	6.3 ± 4.4
③ 2023.1–2024.6	19	1.06 ± 0.78	6.85 ± 3.28	6	5(9)	8	6	1	7.7 ± 3.9	8.9 ± 9.6	15.6 ± 12.5	8.4 ± 4.6
**deep neck**	26	0.25 ± 0.54	5.15 ± 4.43	7	7(13)	7	6	3	13.1 ± 5.5	9.9 ± 8.1	21.3 ± 9.4	14.2 ± 5.5
① 2016.1–2020.6	8	0.15 ± 0.40	4.38 ± 4.69	3	4(4)	1	1	1	15.8 ± 8.7	14.3 ± 6.1	24.4 ± 11.9	17.4 ± 6.3
② 2020.7–2022.12	8	0.27 ± 0.44	5.64 ± 4.62	2	0(3)	6	5	2	13.6 ± 4.8	11.2 ± 11.5	23.1 ± 11.1	14.3 ± 6.2
③ 2023.1–2024.6	10	0.56 ± 0.83	5.29 ± 3.61	2	3(6)	0	0	0	11.6 ± 2.8	8.0 ± 5.0	18.4 ± 5.3	13.0 ± 3.4

GAS: group A *streptococcus pyogenes*, Ag: antigen, iv: intravenous, po: per os.

**Table 2 microorganisms-13-00190-t002:** Bacteria detected from puncture cultures of abscesses are shown.

Pathogen Detected	Total Number	Superficial Cervical	Retropharyngeal	Deep Neck	Peritonsillar
*Fusobacterium* spp.	3			2	1
*Prevottella* spp.	1		1		
*Staphylococcus aureus* (MRSA)	2	2			
*Staphylococcus aureus* (MSSA)	29	27	1	1	
*Streptocuccus pyogenes*	15	4	2		9
*Streptocuccus* spp.	21	12	3	3	3
*Peptostreptococcus* spp.	1				1
Others *	7	3	1		3

*: *Mycobacterium fortuitum, Anaerobic Gram negative and positive bacilli, α-hemolytic streptococci, Neisseria species* (3).

**Table 3 microorganisms-13-00190-t003:** Comparative study of duration of antimicrobial administration and hospitalization with and without puncture drainage.

			Puncture	n	Total Antibiotics Therapy (Days)		Hospital Stay (Days)	
**superficial cervical**								
			yes	48	16.9 ± 9.6	*p* = 0.35	10.7 ± 4.9	*p* < 0.05
			no	35	14.3 ± 2.8		8.9 ± 3.0	
**deep cervical**								
			yes	39	15.0 ± 4.9	*p* = 0.16	8.6 ± 4.0	*p* < 0.01
			no	60	18.6 ± 10.9		12.1 ± 6.0	
**retropharyngeal**								
			yes	12	17.2 ± 4.2	*p* = 0.62	10.4 ± 4.0	*p* = 0.25
			no	18	19.2 ± 6.9		12.9 ± 6.3	
**peritonsillar**								
			yes	20	11.8 ± 2.2	*p* = 0.42	6.2 ± 2.0	*p* < 0.01
			no	27	16.6 ± 12.5		10.0 ± 4.9	
**deep neck**								
			yes	7	20.6 ± 4.3	*p* = 0.43	12.4 ± 3.8	*p* = 0.27
			no	15	21.6 ± 11.0		15.1 ± 5.9	

## Data Availability

The original contributions presented in this study are included in the article/Appendix A. Further inquiries can be directed to the corresponding author.

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
