# Peer review of "Impact of the COVID-19 Pandemic on Epidemiological Trends in Pediatric Cervical Abscess-Forming Infections"

_microorganisms, 2025, doi:10.3390/microorganisms13010190_

Round 1
Reviewer 1 Report
Comments and Suggestions for Authors
This is an interesting observational study with a multicenter approach that investigates the epidemiology of a rare pediatric neck abscess in the Kanto region of Japan from January 2016 to June 2024. This study encompassed a total of 207 cases, constituting an excellent dataset. However, the current draft exhibits several significant issues that are regrettable.
1.Due to the lack of population base data for different years, fluctuations in case numbers do not necessarily reflect alterations in incidence rates. And this is the author's primary research hypothesis or the central issue the authors aims to address.
2. The absence of clear and consistent diagnostic criteria, coupled with a significant number of lost-to-follow-up cases, compromised the validity of the study.
3. It is inadvisable to represent the number of cases in Figure 2 by months rather than years unless there is a clear seasonal distribution pattern of the incidence rate.
4. The representation of the number of cases and infection sites by age group in Figure 3 lacks significance without data on the age distribution of patients seeking medical care.
5. The data in Table 1 is confusing, as some of the information is irrelevant to the study's purpose, such as the use of antibiotics prior to admission, the significant annual variation in the length of hospital stays for patients with deep neck abscesses. However, the absence of comprehensive indications for steroid usage in numerous cases may compromise the reliability of this study concerning case selection and diagnostic accuracy.
6. Regarding the potential factors contributing to the rise in cases following the pandemic, the authors posit that this trend may be associated with an increase in upper respiratory infections, predominantly caused by viral pathogens affecting the respiratory tract. Additionally, the authors note a concurrent increase in cases involving bacterial pathogens responsible for cervical abscesses. However, they are unable to offer a plausible explanation for the interrelation between these two phenomena, such as the mechanism by which viral infections might facilitate subsequent bacterial infections.
7. The discussion on pathogenic factors is challenging to comprehend the author's intended message.
8. "Our study suggests that the M1uk strain carried in the pharynx is unlikely to be associated with a post-pandemic increase in cervical abscess-forming infections," the authors' research lacks sufficient research findings to support such a statement.
Author Response
Dear Reviewer 1
We appreciate your detailed review of our paper and your valuable advices and challenges.
According to your comments, we revised the manuscript, especially expanding the discussion for the bacteriological factor with additional new 16 references cited.
We will respond to each of the points raised.
Corrections or additions are highlighted in the text.
We hope that our responses and corrections are acceptable and will be accepted.
This is an interesting observational study with a multicenter approach that investigates the epidemiology of a rare pediatric neck abscess in the Kanto region of Japan from January 2016 to June 2024. This study encompassed a total of 207 cases, constituting an excellent dataset. However, the current draft exhibits several significant issues that are regrettable.
1.Due to the lack of population base data for different years, fluctuations in case numbers do not necessarily reflect alterations in incidence rates. And this is the author's primary research hypothesis or the central issue the authors aims to address.
âž¡
Thank you for your comment.
We do not see a problem with your point regarding the population ratio, as the participating medical institutions are long-term regional core hospitals and are located in the Tokyo suburbs, where there have been no significant changes in the surrounding demographics over the past 10 years. According to the administrative data (https://www.stat.go.jp/data/jinsui/ accessed on December 26, 2024), the total population of the region where the participating facilities are located increased by a factor of 1.014-1.049 (average 1.032) from 2016 to 2024, while the pediatric population declined by a factor of 0.760-1.075 (average 0.911) during the same period. The average change in the pediatric population over the observational study period is estimated to be 0.911. This rate of change does not affect the results. The effect of fluctuations in the target pediatric population was considered negligible.
- The absence of clear and consistent diagnostic criteria, coupled with a significant number of lost-to-follow-up cases, compromised the validity of the study.
âž¡
Thank you for your comment.
The diagnostic criteria are described in the Materials and Methods section. Although imaging diagnosis by a radiologist or other physician is the standard, we did not have a central diagnostic system, and this is addressed as a problem in our study in the final section of the Discussion.
You are correct that there are about 10% of cases that drop out of follow-up due to transfer to other hospitals. However, we believe that your point is not applicable with regard to epidemiological data on the frequency of occurrence. In addition, there is no chronological bias in the cases transferred to other hospitals, and we believe that the comparative study on the course of treatment is generally valid. Transfer cases were 10 of 101 before the pandemic, 5 of 33 during the pandemic, and 7 of 73 after the pandemic, with no significant differences. The point that some clinical data are missing due to transfer to other hospitals is mentioned as a limitation in the Discussion.
- It is inadvisable to represent the number of cases in Figure 2 by months rather than years unless there is a clear seasonal distribution pattern of the incidence rate.
âž¡
Thank you for your comment.
Initially, we created semi-annual and annual frequency graphs because the frequency of occurrence was so low, but we decided on a monthly graph because the trend we found this time is well reflected in the monthly frequency graph. Furthermore, we decided to use monthly graphs because the monthly graphs are superior to the semiannual graphs to show whether there is seasonality or an epidemic.
- The representation of the number of cases and infection sites by age group in Figure 3 lacks significance without data on the age distribution of patients seeking medical care.
âž¡
Thank you for your comment.
As explained in the text, the graph is significant in that it shows that superficial abscess-forming infections (primarily suppurative lymphadenitis) are more common in young children, while deep abscess-forming infections span a wide age range. We believe that it also makes the data more understandable to the reader in that it complements the age data for morbidity in Table 1. As mentioned in your question 1, according to the administrative data (https://www.stat.go.jp/data/jinsui/ accessed on December 26, 2024), the total population of the region where the participating facilities are located increased by a factor of 1.014-1.049 (average 1.032) from 2016 to 2024, while the pediatric population declined by a factor of 0.760-1.075 (average 0.911) during the same period. Cervical abscess-forming infections are treated in the hospital and should be considered to reflect the frequency of occurrence.
- The data in Table 1 is confusing, as some of the information is irrelevant to the study's purpose, such as the use of antibiotics prior to admission, the significant annual variation in the length of hospital stays for patients with deep neck abscesses. However, the absence of comprehensive indications for steroid usage in numerous cases may compromise the reliability of this study concerning case selection and diagnostic accuracy.
âž¡
Thank you for your comment.
We believe that this is a significant study item in that it shows that there is no significant difference in the incidence of cervical abscess-forming infections before and after the pandemic. Of course, it is not enough to discuss the issue of whether there is a relationship between the frequency of antimicrobial use and the frequency of cervical abscess-forming infections, but we believe that differences in the frequency of antimicrobial use, although limited, at least unlikely have an impact on the frequency of cervical abscess-forming infections.
We believe that a very detailed index would be needed to compare differences in disease severity. Given that this is a retrospective study, we believe that the most objective measure of disease severity is the duration of hospitalization, which we have included in Table 1.
Similarly, we included steroid use as a survey item because we believe it may be one indicator of disease severity. We believe that the frequency of steroid use is not as high as we had expected. It is rather important for our observational study to show that there is no significant difference in the frequency of steroid use before and after the pandemic.
- Regarding the potential factors contributing to the rise in cases following the pandemic, the authors posit that this trend may be associated with an increase in upper respiratory infections, predominantly caused by viral pathogens affecting the respiratory tract. Additionally, the authors note a concurrent increase in cases involving bacterial pathogens responsible for cervical abscesses. However, they are unable to offer a plausible explanation for the interrelation between these two phenomena, such as the mechanism by which viral infections might facilitate subsequent bacterial infections.
âž¡
Thank you for your comments.
This observation was a surprise to us as well.
Initially, we thought that the major factors were the decrease in immune function in children due to the drastic decrease in infectious diseases in the pandemic or the emergence of virulent bacteria. However, our observations did not support the above hypothesis.
There was little mention of cervical abscess-forming infections being associated with common respiratory viral infections. Although we cannot determine a detailed causal relationship, we believe that our observations are rather significant, at least in that we can say that there is a profound relationship between respiratory viral infections and cervical abscess-forming infections, although only from an epidemiological association. The association with viral infection is not asserted definitively, but only stated as a probability based on observations.
- The discussion on pathogenic factors is challenging to comprehend the author's intended message.
âž¡
Thank you for your comments.
In recent years, there has been no discussion on the etiology and pathogenesis of cervical abscess-forming infections, and we hope that our observations will serve as one suggestion and lead to new research and other claims that we hope will be made. In our present study, we did not have sufficient data to examine bacteriological factors due in part to the fact that our study was retrospective.
This problem is mentioned in the limitation section and will be addressed in future studies.
- "Our study suggests that the M1uk strain carried in the pharynx is unlikely to be associated with a post-pandemic increase in cervical abscess-forming infections," the authors' research lacks sufficient research findings to support such a statement.
âž¡
Thank you for your comments.
As discussed in the text, an increase in invasive streptococcal infections in adults and children and an increase in detected M1uK strains have been noted in Japan. However, epidemiological data on the organisms responsible for streptococcal pharyngitis from official surveillance in Tokyo do not show an increase in the T1 serotype to which M1uk strains belong. Furthermore, no increase in streptococci was observed in bacteria detected in puncture cultures of cases in an observational study. Based on the above indirect and direct data, we believe that our assertion has sufficient validity. However, as you said, there is no hard evidence, so I changed the wording slightly. “Further detailed epidemiological and bacteriological analysis is needed to determine the involvement of M1uk strains in the increase in cervical abscess-forming infections.”
Submission Date
05 December 2024
Date of this review
18 Dec 2024 15:29:37
Reviewer 2 Report
Comments and Suggestions for Authors
Thank you for inviting me to review this manuscript. It is interesting and well-written. I have some comments that could be of use:
1. Please re-organize the abstract section in one paragraph. No need to change paragraphs every time a subsection changes
2. Line 190: P. pyogenes? Maybe S. pyogenes?
3. Lines 190, 195, Table 2, and elsewhere: All names of microorganisms (gender and species) should be in italics throughout the manuscript
4. Line 197: Do you mean Panton-Valentine leukocidin? This is a more recognizable term
5. Line 203, 205 and elsewhere: All genera should start with a capital letter
6. Data availability statement: This should be corrected
7. Table 1: Please add a footnote where all the abbreviations will be explained in full (GAS, iv, po)
8. Table 1 and Table 3: please improve the table in terms of structure, lining etc
9. Table 2: a heading for the first column of numbers is missing. I guess it is ‘Total’?
10. Table 2: Please add a footnote describing what ‘other’ microorganisms were isolated
11. Line 208: Please delete this ‘As mentioned in the Introduction’
12. Line 220: Our study suggests that the M1uk strain carried in the pharynx is unlikely to be associated with a post-pandemic increase in cervical abscess-forming infections à I am skeptical about this. It would be more convincing to show a table showing the temporal distribution of isolated pathogens. A stable rate of identification of streptococci would be convincing enough to support this sentence
13. Line 231-241: This is also difficult to support without providing data on antimicrobial resistance. If the authors have such data, they could provide them as supplementary material
14. Line 244: the rate of blood culture positivity is not mentioned in the results section, neither is the proportion of patients who had blood cultures taken
15. Line 250: Replace ‘second’ with another word (it is not the second limitation mentioned)
16. Line 254: Those patients who were transferred could also have more severe disease, and this could be associated with introduction of bias, microbiologically speaking
17. Line 260: Replace ‘pathology’ with ‘pathophysiology’
18. Line 261: replace ‘pandemic’ with ‘COVID-19 pandemic’
Author Response
Dear Reviewer 2
We appreciate your detailed review of our paper and your valuable advices and challenges.
According to your comments, we revised the manuscript, especially expanding the discussion for the bacteriological factor with additional new 16 references cited.
We will respond to each of the points raised.
Corrections or additions are highlighted in the text.
We hope that our responses and corrections are acceptable and will be accepted.
Thank you for inviting me to review this manuscript. It is interesting and well-written. I have some comments that could be of use:
- Please re-organize the abstract section in one paragraph. No need to change paragraphs every time a subsection changes
âž¡
Thank you for your comments.
I have corrected it as you pointed out.
- Line 190: P. pyogenes? Maybe S. pyogenes?
âž¡
Thank you for your comments.
I have corrected it as you pointed out.
- Lines 190, 195, Table 2, and elsewhere: All names of microorganisms (gender and species) should be in italics throughout the manuscript
âž¡
Thank you for your comments.
I have corrected it as you pointed out.
- Line 197: Do you mean Panton-Valentine leukocidin? This is a more recognizable term
âž¡
Thank you for your comments.
I have corrected it as you pointed out.
- Line 203, 205 and elsewhere: All genera should start with a capital letter
âž¡
Thank you for your comments.
I have corrected it as you pointed out.
- Data availability statement: This should be corrected
âž¡
Thank you for your comments.
I have corrected it as you pointed out.
“Data available on request from the authors. The raw data supporting the conclusions of this article will be made available by the authors on request.”
- Table 1: Please add a footnote where all the abbreviations will be explained in full (GAS, iv, po)
âž¡
Thank you for your comments.
I have corrected it as you pointed out.
- Table 1 and Table 3: please improve the table in terms of structure, lining etc
âž¡
I will ask the Printing Office.
- Table 2: a heading for the first column of numbers is missing. I guess it is ‘Total’?
âž¡
Thank you for your comments.
I have corrected it as you pointed out.
- Table 2: Please add a footnote describing what ‘other’ microorganisms were isolated
âž¡
Thank you for your comments.
I have corrected it as you pointed out.
* : Mycobacterium fortuitum, Anaerobic Gram negative and positive bacilli, α-hemolytic streptococci, Neisseria species (3)
- Line 208: Please delete this ‘As mentioned in the Introduction’
âž¡
Thank you for your comments.
I have corrected it as you pointed out.
- Line 220: Our study suggests that the M1uk strain carried in the pharynx is unlikely to be associated with a post-pandemic increase in cervical abscess-forming infections à I am skeptical about this. It would be more convincing to show a table showing the temporal distribution of isolated pathogens. A stable rate of identification of streptococci would be convincing enough to support this sentence
âž¡
Thank you for your comments.
As discussed in the text, an increase in invasive streptococcal infections in adults and children and an increase in detected M1uK strains have been noted in Japan. However, epidemiological data on the organisms responsible for streptococcal pharyngitis from official surveillance in Tokyo do not show an increase in the T1 serotype to which M1uk strains belong. Furthermore, no increase in streptococci was observed in bacteria detected in puncture cultures of cases in an observational study. Based on the above indirect and direct data, we believe that our assertion has sufficient validity. However, as you said, there is no hard evidence, so I changed the wording slightly. “Further detailed epidemiological and bacteriological analysis is needed to determine the involvement of M1uk strains in the increase in cervical abscess-forming infections.”
- Line 231-241: This is also difficult to support without providing data on antimicrobial resistance. If the authors have such data, they could provide them as supplementary material
âž¡
It is difficult to come up with an accurate reference for antimicrobial resistance in potentially originating bacteria related to this study. We have added a description of the transition of resistance of staphylococcus aureus in Japan as a whole. “According to JANIS (Japan Nosocomial Infections Surveillance) data (https://janis.mhlw.go.jp/report/kensa.html, accessed on December 28, 2024), from 2018 to 2023, the frequency of MRSA among all Staphylococcus aureus isolated from inpatients decreased from 47.8% to 44.0%, and the frequency of MRSA among all Staphylococcus aureus isolated from outpatients decreased from 30.1% to 29.3%. It is unlikely that the rate of staphylococcal resistance is increasing in Japan as a whole.”
- Line 244: the rate of blood culture positivity is not mentioned in the results section, neither is the proportion of patients who had blood cultures taken
âž¡
We added the information as below.
“Blood cultures were performed in 94 of 96 patients with superficial neck abscesses and 82 of 111 patients with deep neck abscesses, none of which were positive.”
- Line 250: Replace ‘second’ with another word (it is not the second limitation mentioned)
âž¡
Thank you for your comments.
I have corrected it as you pointed out.
- Line 254: Those patients who were transferred could also have more severe disease, and this could be associated with introduction of bias, microbiologically speaking
âž¡
I think it is as you pointed out.
Transfer cases were 10 of 101 before the pandemic, 5 of 33 during the pandemic, and 7 of 73 after the pandemic, with no significant differences in the three periods. Therefore, we do not believe it will have a significant impact on our interpretation.
- Line 260: Replace ‘pathology’ with ‘pathophysiology’
âž¡
Thank you for your comments.
I have corrected it as you pointed out.
- Line 261: replace ‘pandemic’ with ‘COVID-19 pandemic’
âž¡
Thank you for your comments.
I have corrected it as you pointed out.
Submission Date
05 December 2024
Date of this review
13 Dec 2024 09:07:32
Round 2
Reviewer 1 Report
Comments and Suggestions for Authors
My primary concern regarding this manuscript is still the absence of standardized diagnostic criteria across multiple centers. The authors relied on diagnoses extracted from the medical history information system, which may vary depending on the physician's expertise. Additionally, several clinical details are unclear, such as the rationale for performing enhanced CT scans on all children, the reasons for the low and negative rate of blood cultures, the indications for steroid use, duration of antibiotic use and hospital stay could not available in most of the cases, and the lack of differentiation in antibiotic therapy duration between superficial and deep cervical abscesses. The authors need to address these in the discussion section.
Furthermore, I would recommend that the authors reconsider the presentation approach of the content. For instance, the discussion section should systematically address the results of the main study in detail, particularly the data presented in the tables. Additionally, the conclusions drawn in the abstract should accurately reflect and correspond to the reported findings.

Author Response
Comments of Review 1:
My primary concern regarding this manuscript is still the absence of standardized diagnostic criteria across multiple centers. The authors relied on diagnoses extracted from the medical history information system, which may vary depending on the physician's expertise. Additionally, several clinical details are unclear, such as the rationale for performing enhanced CT scans on all children, the reasons for the low and negative rate of blood cultures, the indications for steroid use, duration of antibiotic use and hospital stay could not available in most of the cases, and the lack of differentiation in antibiotic therapy duration between superficial and deep cervical abscesses. The authors need to address these in the discussion section.
Furthermore, I would recommend that the authors reconsider the presentation approach of the content. For instance, the discussion section should systematically address the results of the main study in detail, particularly the data presented in the tables. Additionally, the conclusions drawn in the abstract should accurately reflect and correspond to the reported findings.
Dear Reviewer 1,
Thank you very much for your valuable comments and suggestions.
We have revised the paper according to your comments with five new references added. The corrected parts are highlighted in green. (The first revision is highlighted in yellow.)
We hope that this revision resolves your comments.
This study is a retrospective study and has not been able to define a standardized diagnostic process strictly. This point is mentioned in LIMITATION. In accordance with the general practice guidelines (ref Children 2022, 9, 618.), each facility actively performed contrast-enhanced CT scans on cases with Red Flag signs such as neck pain and dysphagia to advance the diagnosis (line 84-86). As mentioned in various review articles, the golden standard for the diagnosis of cervical abscess is said to be contrast-enhanced CT findings. For final determination, contrast-enhanced CT findings in the medical information were judged and confirmed by physicians and radiologists at each facility in this study. We think there were no problem with the diagnostic process. Additional references have been added for treatment procedures and other information (line94-100).
We added that no cases required emergency airway clearance (line 149-150).
We added description of when puncture drainage was performed (line 225-228).
A discussion of the appropriate duration of antimicrobial therapy was added (line 364-370).
A discussion of blood culture positivity rates was added with references (line 371-378).
In the summary, there was an insufficient mention of the association with respiratory viral infections, so this has been corrected and added (line 41, line 49-50).
Reviewer 2 Report
Comments and Suggestions for Authors
The manuscript has been improved now.
Author Response
Dear Reviewer 2
Thank you for your evaluation of my revised paper.
We have made some corrections as noted by other reviewers and hope you will review them.
The revised parts are listed below.
The corrected parts are highlighted in green. (The first revision is highlighted in yellow.)
This study is a retrospective study and has not been able to define a standardized diagnostic process strictly. This point is mentioned in LIMITATION. In accordance with the general practice guidelines (ref Children 2022, 9, 618.), each facility actively performed contrast-enhanced CT scans on cases with Red Flag signs such as neck pain and dysphagia to advance the diagnosis (line 84-86). As mentioned in various review articles, the golden standard for the diagnosis of cervical abscess is said to be contrast-enhanced CT findings. For final determination, contrast-enhanced CT findings in the medical information were judged and confirmed by physicians and radiologists at each facility in this study. We think there were no problem with the diagnostic process. Additional references have been added for treatment procedures and other information (line94-100).
We added that no cases required emergency airway clearance (line 149-150).
We added description of when puncture drainage was performed (line 225-228).
A discussion of the appropriate duration of antimicrobial therapy was added (line 364-370).
A discussion of blood culture positivity rates was added with references (line 371-378).
In the summary, there was an insufficient mention of the association with respiratory viral infections, so this has been corrected and added (line 41, line 49-50).